# Quercetin enhances motility in aged and heat-stressed *Caenorhabditis elegans* nematodes by modulating both HSF-1 activity, and insulin-like and p38-MAPK signalling

**Takaya Sugawara** , **Kazuichi Sakamoto** *

Graduate School of Life and Environmental Sciences, University of Tsukuba, Tsukuba, Ibaraki, Japan

* sakamoto@biol.tsukuba.ac.jp

**Data Availability Statement:** All relevant data are within the manuscript and its Supporting Information files.

## Abstract

Quercetin is a yellow pigment that is found in many common dietary plants, and that protects against oxidative stress, inflammation, and arteriosclerosis. It has also been suggested to prolong the lifespan of, and enhance heat-stress tolerance in nematodes; thus, the present study investigated its effects on both the nematode life- and health span by assessing its capacity to promote nematode motility after aging and/or heat stress, as well as the mechanisms underlying these effects. The results of the conducted analyses showed that quercetin feeding prolonged lifespan, suppressed age-related motility retardation, improved motility recovery after heat stress, and decreased the production of both intercellular and mitochondrial reactive oxygen species in the analysed *Caenorhabditis elegans* strains, likely by modulating the insulin-like signalling (ILS) pathway and p38-mitogen-activated protein kinase (MAPK) pathway. In particular, the transcription factors DAF-16 and SKN-1 were found to mediate the observed quercetin-induced effects, consistent with their previously demonstrated roles as regulators of aging. Furthermore, we demonstrated, for the first time, that quercetin induced heat-stress tolerance in *C. elegans* by modulating HSF-1 expression and/or activity. Thus, the present study provides valuable insights into the mechanisms by which quercetin inhibit aging and enhance heat-stress tolerance via ILS and MAPK pathway in *C. elegans*.

## Introduction

Recent studies have identified a range of biologically active substances in foods; for example, polyphenols such as resveratrol and anthocyanin have been detected in the grape pericarp, and shown to exert strong anti-oxidative effects, and to reduce the risk of arteriosclerosis and stroke [1–3]. Quercetin used in this study is contained in many different of dietary plants and classified as a flavonoid, one kind of polyphenol. So far, various flavonoids such as catechin and anthocyanin have been demonstrated to incur physiological effects in many kinds of model organism [4,5]. The yellow flavonoid quercetin and its glycoside have been reported to

**Funding:** We clarify that this work was supported in part by Suntory Global Innovation Center Limited and Grants-in-Aid for Scientific Research and Education from the University of Tsukuba, Japan. We declare as follows; "The funders had no role in study design, data collection and analysis, decision to publish, or preparation of the manuscript." There was no additional external funding received for this study.

**Competing interests:** The authors have declared that no competing interests exist.

inhibit oxidative stress, inflammation, and cerebrovascular disease, tumour, and cancer development [6–12]. To date, the effects of quercetin on lifespan and protection against stress have been researched in some model organisms [13–17]. The physiological effects of quercetin should be different depending on the kinds of model organisms and concentrations [13]. The effects on health span have not reported; thus, the present study aimed to analyse the physiological effects of quercetin on 'health span' in *C. elegans*. Health span is the period for which an individual remains healthy and independent; for example, the period for which no extensive medical care or nursing is required in humans, whereas we defined health span as the period maintaining the normal locomotion in *C. elegans*.

*C. elegans* is a nematode with a transparent body that makes thrashing movements. It can be easily cultured on an agar medium at room temperature (20˚C), and is widely used as a model organism in a wide range of biological studies due to the fact that it shares orthologous genes and various basic organs (e.g. digestive and nervous systems) with higher order animals. In addition, *C. elegans* has a short lifespan that allows many assays to be conducted in a limited period of time. Many studies have already been conducted to evaluate the effect of functional dietary components on aging and stress tolerance in *C. elegans* [18–20]; thus, it is a very suitable model to use to conduct preliminary assessments of physiological active compounds, and thereby provide valuable and relevant data to support subsequent studies in higher order animals.

Previous studies have suggested that quercetin may prolong the lifespan and increase stress tolerance in *C. elegans* [4,16,17,20]; however, its effects on motility are poorly understood in general, and have not yet been assessed in aged or stressed nematodes [21]. Therefore, the present study aimed to elucidate the physiological effects of, and mechanisms by which quercetin impacts motility in aged and heat-stressed *C. elegans* nematodes. Furthermore, while previous studies have suggested that heat stress tolerance can be induced in nematodes by modulating ILS and p38-mitogen-activated protein kinase (MAPK) pathway [4,15–17], the present study aimed to further clarify the signalling pathways by which quercetin modulates *C. elegans* aging and heat-stress tolerance.

## Materials and methods

### Nematodes and quercetin

The *C. elegans* N2 Bristol (wild type), *daf-2(e1370)*, *age-1 (hx546)*, *daf-16 (mgDf50)*, *sek-1 (ag1)*, *hsf-1 (sy441)* (Caenorhabditis Genetics Center (CGC), University of Minnesota, Minneapolis, MN, USA), *nsy-1 (tm850)*, *pmk-1 (tm676)*, and *skn-1 (tm4241)* (National BioResource Project (NBRP), Tokyo Women's Medical University, Japan) nematode strains were cultured (20˚C) on nematode growth medium (NGM) plates that were spread with *Escherichia coli* (OP50) (Table 1) [22,23]. Quercetin (Cayman chemical company, Michigan, USA) was dissolved in dimethyl sulfoxide (DMSO; Kanto Chemical Co., Inc., Tokyo, Japan) to a concentration of 0.01 M, before being diluted to 50- and 500 μM in OP50, and spread onto NGM plates for each quercetin-feeding assay. We defined the plates spread onto NGM plates with OP50 as OP plate and with quercetin, containing OP50 as Q plate. 200 μl of OP50 and quercetin were spread onto NGM plates and dried.

### Nematode synchronization

To collect synchronized eggs, adult nematodes were crushed in NaClO solution comprising (1:10) 10 N NaOH (Wako Pure Chemical Industries, Ltd., Osaka, Japan) in NaClO (Haiter, KAO, Tokyo, Japan).

**Table 1. Genes shown in this study.**

| Genes | Properties |
|---|---|
| *daf-2* | Encodes receptor for insulin-like ligands. Gene deficient mutant shows extended lifespan. |
| *age-1* | Encodes an ILS pathway protein homologous to PI3K. Gene deficient mutant shows extended lifespan. |
| *daf-16* | Encodes a FOXO-homologue transcription factor that is regulated by ILS, and that mediates aging and stress tolerance. |
| *nsy-1* | Encodes a p38-mitogen-activated protein kinase kinase kinase (MAPKKK) protein that acts in the MAPK cascade. |
| *sek-1* | Encodes a MAPKK protein that acts in the MAPK cascade. |
| *pmk-1* | Encodes a MAPK protein that acts in the MAPK cascade. |
| *skn-1* | Encodes a transcription factor homologous to Nrf2 that is regulated by MAPK, and that mediates aging and stress tolerance. |
| *hsf-1* | Encodes a transcription factor that regulates heat-stress tolerance. |
| *sod-1* | SKN-1 target gene that encodes the reactive oxygen species (ROS) scavenger, SOD-1. |
| *sod-2* | SKN-1 target gene that encodes the ROS scavenger SOD-2. |
| *sod-3* | DAF-16 target gene that encodes the ROS scavenger SOD-3. |
| *ctl-1* | DAF-16 target gene that encodes CTL-1 that metabolizes $H_2O_2$ to $H_2O$. |
| *ctl-2* | DAF-16 target gene that encodes CTL-2 that metabolizes $H_2O_2$ to $H_2O$. |
| *hsp-12.6* | Gene regulated by both DAF-16 and HSF-1 that encodes a heat shock protein (HSP). |
| *hsp-16.1* | Gene regulated by both DAF-16 and HSF-1 that encodes an HSP. |
| *hsp-16.2* | HSF-1 target gene that encodes an HSP. |
| *hsp-70* | HSF-1 target gene that encodes an HSP. |
| *sip-1* | Gene that is regulated by both DAF-16 and HSF-1, and that mediates heat-stress tolerance. |
| *mtl-1* | DAF-16 target gene that mediates heat-stress tolerance. |

## Evaluation of lifespan

Synchronized worms were cultured on OP plates at 20˚C for 96 h, and then transferred to OP or Q plates (50 μM and 500 μM, day 0) for further culture (20˚C). Offspring generation was prevented via the addition of 0.5 mg/ml of 2'-deoxy-5-fluorouridine (FUdR; Wako Pure Chemical Industries, Ltd.) to the plates on days -1, 0, and 2. The worms were transferred to new plates every 2 days, and the survival rate of the wild type worms were counted on each transfer day, where the survival rate at day 0 was set as 100%. Worms that displayed no movement upon gentle probing with a platinum picker were judged as dead. The survival rate of CT (dDW), containing double distilled water (dDW) as solvent instead of DMSO was also counted. Assay was conducted at least three times independently.

## Evaluation of motility in ageing nematodes

Synchronized worms were cultured on OP plates at 20˚C for 96 h, and then transferred to OP or Q plates (50 μM and 500 μM, day 0) for further culture (20˚C). Offspring generation was prevented via the addition of 0.5 mg/ml of 2'-deoxy-5-fluorouridine (FUdR; Wako Pure Chemical Industries, Ltd.) to the plates on days -1, 0, and 3. The worms were transferred to new plates every 3 days, and the thrashing movements in 15 seconds of the wild type, *daf-2 (e1370)*, *age-1(hx546)*, *daf-16(mgDf50)*, *nsy-1(tm850)*, *sek-1(ag1)*, *pmk-1(tm676)*, *skn-1 (tm4241)*, and *hsf-1(sy441)* worms were counted on each transfer day in S-basal medium, where the movement at day 0 was set as 100%. The thrashing movements of CT (dDW), containing dDW as solvent instead of DMSO was also counted. The movement of the *daf-2*

(*e1370)* and *age-1(hx546)* worms was counted until day 12 due to their extended lifespan [22]. All assays were conducted at least three times independently.

## Evaluation of nematode motility recovery after heat stress

Generally, heat stress decreases movement of worms. To measure the motility-recovery rate after heat stress, synchronized wild type, *daf-2(e1370)*, *age-1(hx546)*, *daf-16(mgDf50)*, *nsy-1 (tm850)*, *sek-1(ag1)*, *pmk-1(tm676)*, *skn-1(tm4241)*, and *hsf-1(sy441)* worms were cultured on OP or Q plates (50 μM and 500 μM) at 20°C for 96 h. In order to prevent the growth of *E. coli* from affecting worms, worms were transferred to NGM plates without spread of OP50, and then incubated at 35°C for 4 h. Heat-stress initiation was designated as 0 h, and the thrashing movements in 15 seconds of both heat-stressed and control (maintained at 20°C) worms were counted every 6 or 12 h in S-basal medium. The ratio of the movement counts generated for the heat-stressed and control worms (n = 10 worms/group) was calculated [24]. The motility-recovery rate of CT (dDW), containing dDW as solvent instead of DMSO was also counted in wild type. The movement of the *daf-16(mgDf50)* and *hsf-1(sy441)* worms did not recover sufficiently at 12 h; therefore, they were evaluated for 24 h. All assays were conducted at least three times independently.

## Evaluation of mitochondrial ROS

Synchronized worms were cultured on OP or Q plates (50 μM and 500 μM) for 72 h, before 400 μL of 0.5 mg/mL MitoTracker® Orange CM-H2TMRos (Thermo Fisher Scientific, Inc., Waltham, MA, USA) was added to the plates. After 24 h, the worms were washed, and fixed in 10% ethanol (Kanto Chemical Co.). Fluorescence (n > 30 worms/group) was measured using a BZ8000 microscope, and analysed using ImageJ software (NIH). The fluorescence level exhibited by control worms was set as 100%. Assays were conducted at least three times independently.

## Evaluation of cellular ROS

Synchronized worms were cultured on OP or Q plates (50 μM and 500 μM) for 96 h, before DCF-DA (Wako) was diluted in dimethyl sulfoxide (Kanto Chemical Co.) to produce a 50 μM DCF-DA solution that was added (400 μL) to each plate. After 1 h, worms were washed, and fixed in 10% ethanol. Again, fluorescence (n > 35 worms/group) was measured using a BZ8000 microscope, and analysed using ImageJ software. The fluorescence level exhibited by control worms was set as 100%. Assays were conducted at least three times independently.

## Gene expression

Synchronized worms were cultured on OP or Q plates (50 μM and 500 μM) for 96 h, before mRNA for genes of interest (Table 1) [25–30] was extracted via crushing using the Power Masher® and Bio Masher® (Nippi Inc., Tokyo, Japan), and reverse-transcribed to cDNA using the PrimeScript™ RT Reagent Kit and the gDNA Eraser (Takara, Shiga, Japan). A real-time quantitative PCR (qPCR) was conducted using the Thermal Cycler Dice® Real Time System Lite (Takara) with Thunderbird® SYBR® qPCR Mix (TOYOBO, Osaka, Japan), and appropriate primers (Table 2). Actin was selected as a reference gene and confirmed the suitability by comparing with *tba-1* and *pmp-3*, used as alternative reference genes [31]. Each qPCR reaction was performed in triplicate. Assays were conducted at least three times independently.

**Table 2. Primer sequences used in the conducted gene-expression analysis.**

| Gene | Sense primer (5'–3') | Antisense primer (5'–3') |
|---|---|---|
| actin[*] | TCGGTATGGGACAGAAGGAC | CATCCCAGTTGGTGACGATA |
| daf-16 | ATCATCTTTCCGTCCCCG | TTGGAATTGCTGGAACCG |
| skn-1 | AGTGTCGGCGTTCCAGATTTC | GTCGACGAATCTTGCGAATCA |
| hsf-1 | ATGTACGGCTTCCGAAAGATGA | TCTTGCCGATTGCTTTCTCTTAA |
| sod-1 | CGTAGGCGATCTAGGAAATGTG | TGACGAGCGTGTCGGTGAG |
| sod-2 | GATACTGTCCAAAGGGAAAGAT | GTAGTAAGCGTGCTCCCAGA |
| sod-3 | ATCTACTGCTCGCACTGCTT | TTTCATGGCTGATTACAGGTT |
| ctl-1 | CGATACCGTACTCGTGATGAT | CCAAACAGCCACCCAAATCA |
| ctl-2 | CTGGGAGAAGGTGTTGGAT | GGATGAACCTTTGAAAAGTGAT |
| hsp-12.6 | TGGAGTTGTCAATGTCCTCG | GACTTCAATCTCTTTTGGGAGG |
| hsp-16.1 | CCACTATTTCCGTCCAGCTCA | GGCGCTTGCTGAATTGGAAT |
| hsp-16.2 | TGTTGGTGCAGTTGCTTCGAATC | TTCTCTTCGACGATTGCCTGTTG |
| hsp-70 | ACCCTTCGTTGGATGGAACG | GCATCCGGAACCTGATTGGGC |
| sip-1 | CGAGCACGGGTTCAGCAAGAG | CAGCGTGTCCAGCAGAAGTGTG |
| mtl-1 | TGTGAGGAGGCCAGTGAGA | TTAATGAGCCGCAGCAGTT |

[*]Reference gene.

## Statistical analysis

Data were presented as the mean ± SEM, and statistical analysis was performed using one way ANOVA followed by *Tukey's HSD* post hoc test. The survival rate was analysed using the *log-rank test*. Graphs were generated using Microsoft Excel and Microsoft PowerPoint software (Microsoft Corp., Redmond, WA, USA). A P value < 0.05 was considered to indicate statistical significance.

## Results

Previous studies have reported that quercetin can prolong the nematode lifespan [4,15–17,20]. At first, we investigated whether DMSO used as solvent has toxicity to the lifespan of nematode and whether 50- and 500μM quercetin used in present study can induce longevity effect to nematode, thereby we designed two controls. CT (dDW) contains the same amount of double distilled water (dDW) with DMSO contained in CT (DMSO), 50- and 500 μM quercetin samples. The survival rate of CT (dDW) and CT(DMSO) didn't change significantly and 50- or 500 μM quercetin was increased in wild type (Fig 1). These results indicate the concentration of DMSO used in this study don't affect the longevity of nematode, and 50- and 500 μM quercetin can prolong the lifespan similarly to the previous study [4,15–17,20]. Present study focused on analysis of its effects on the nematode health span, rather than longevity. Specifically, we analysed the effects of quercetin on nematode motility through ageing process by feeding nematodes 50- or 500 μM quercetin and evaluating their movement every 3 days. Motility deterioration was suppressed on days 3,9 and 12 in wild-type worms that were fed 50 μM quercetin, and on days 3, 6, 9 and 12 in wild-type worms that were fed 500 μM quercetin (Fig 2A). In order to find differences with other flavonoids, we used epigallocatechin gallate (EGCG), a species of catechin [5]. EGCG showed similar effect with 500 μM of quercetin (S1 Fig). Suppression of motility deterioration was observed in the *hsf-1(sy441)* worms similarly (Fig 2I); however, in contrast, quercetin feeding did not suppress motility deterioration in the *daf-2(e1370)*, *age-1(hx546)*, *daf-16(mgDf50)*, *nsy-1(tm850)*, *sek-1(ag1)*, *pmk-1(tm676)*, nor the

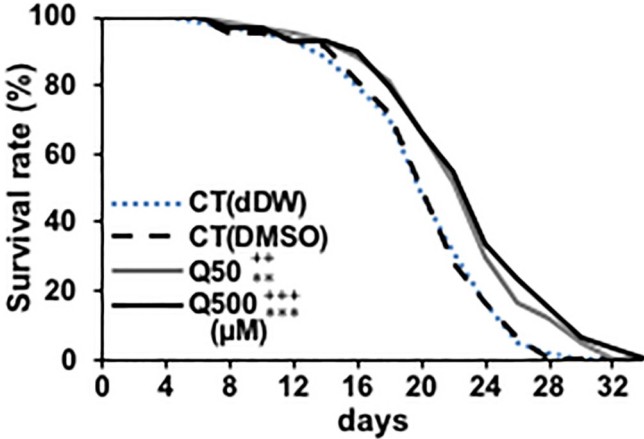

**Fig 1. Quercetin induced life span extension.** Survival rate analysis was generated for N2 wild-type *Caenorhabditis elegans* nematodes every 2 days and presented relative to that calculated on day 0. n = 60/group. ++P < 0.01, +++P < 0.005 vs CT (dDW) and **P < 0.01, ***P < 0.005 vs CT (DMSO) according to the conducted *log-rank test*. All assays were conducted at least three times independently.

*skn-1*(*tm4241*) nematodes (Fig 2B–2H), suggesting that it may promote motility by modulating *daf-2*, *age-1*, *daf-16*, *nsy-1*, *sek-1*, *pmk-1* and *skn-1*, but not *hsf-1* expression and/or activity. Similar to lifespan analysis, the motility rate of CT (dDW) and CT(DMSO) didn't change significantly (Fig 2A).

Upregulated stress tolerance has been shown to be closely related to longevity and anti-aging effects in nematodes [19], and previous studies have revealed that quercetin improves nematode heat-stress tolerance [16,20]. Thus, the present study analysed the motility of nematodes after heat-stress induction, with and without quercetin feeding. The motility-recovery rate of the N2 wild-type worms that were fed with 50- and 500 μM quercetin was resultantly shown to be non-significantly and significantly increased compared to controls, respectively (Fig 3A). The thrashing recovery at hours 24 was didn't change significantly, due to the full recovery from heat damages (Fig 3A). The recovery of motility was shown in the worms fed EGCG similarly to the worms fed 500 μM of quercetin (S2 Fig). In contrast, the recovery rate of the *daf-2*(*e1370*), *age-1*(*hx546*), *nsy-1*(*tm850*), *sek-1*(*ag1*), *pmk-1*(*tm676*), *skn-1*(*tm4241*) and *hsf-1*(*sy441*) nematodes after heat-stress induction was unchanged by quercetin feeding (Fig 3B, 3C and 3E–3I). This suggests that quercetin may promote motility-recovery by modulating *daf-2*, *age-1*, *daf-16*, *nsy-1*, *sek-1*, *pmk-1*, *skn-1* and *hsf-1* expression and/or activity. Notably, the motility of the *daf-16*(*mgDf50*) worms that were fed with quercetin after heat stress was partially recovered (Fig 3D), suggesting that motility-recovery after heat stress may occur via incurred effects on *daf-16* activity.

Heat stress has been shown to cause nematodes to produce reactive oxygen species (ROS), which are associated with a range of biological effects, including cell damage, and disease [32]. The mitochondria are a major source of ROS, particularly during ATP generation. Since quercetin improved the motility of the ageing nematodes, we expected that the mitochondria activated during this process would produce increased ROS levels (Fig 2A). Surprisingly, however, mitochondrial and cellular ROS levels were significantly decreased after quercetin feeding in the analysed nematodes (Fig 4A and 4B). In order to clarify the mechanism and to identify genes involved in the observed quercetin-induced physiological effects, qPCR analysis was conducted. We selected actin as a reference gene in this study and it could be appropriate

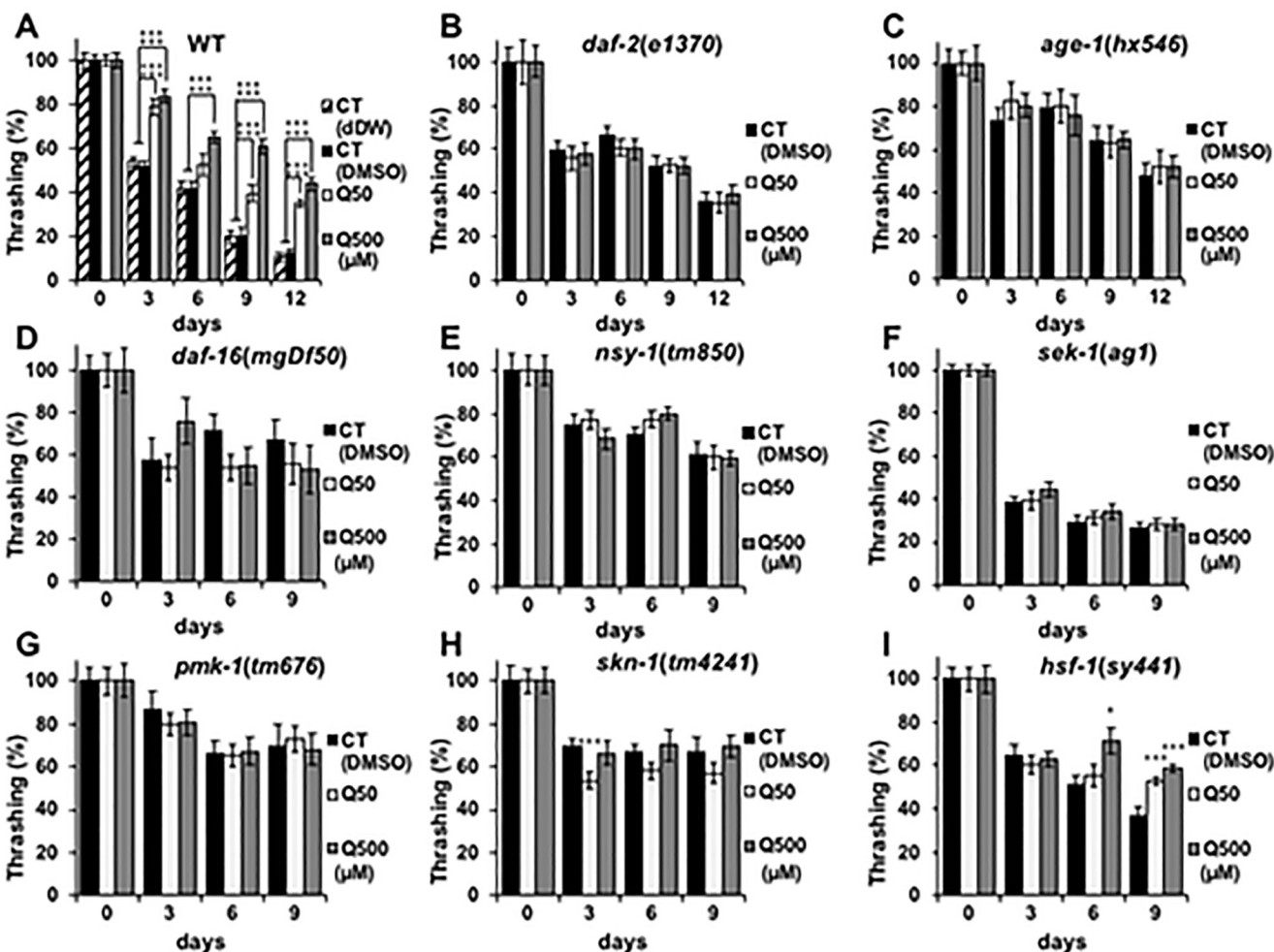

**Fig 2. Quercetin suppressed age-related retardation.** Movement counts were generated for **(A)** N2, **(B)** *e1370*, **(C)** *hx546*, **(D)** *mgDf50*, **(E)** *tm850*, **(F)** *ag1*, **(G)** *tm676*, **(H)** *tm4241*, and **(I)** *sy441 Caenorhabditis elegans* nematodes every 3 days from day 0 until **(A, B, E–I)** day 9 or **(C, D)** day 12, and presented relative to that calculated on day 0. Data are presented as the mean ± SEM, n = 10/group. +++P < 0.005 vs CT (dDW) and *P < 0.05, ***P < 0.005 vs CT (DMSO) according to the conducted *Tukey's HSD*. All assays were conducted at least three times independently.

(S3 Fig), because the differences of quantities of actin and other alternative reference genes, *tba-1* and *pmp-3*, in each condition are not significant [31]. We chose some genes mainly related to three transcriptions factors, DAF-16, SKN-1 and HSF-1. Genes chosen in present study were specifically related to stress tolerance (Table 1). Feeding with either 50- or 500 μM quercetin was resultantly shown to significantly upregulate *daf-16*, *skn-1*, *hsf-1*, *sod-1*, *sod-2*, *ctl-1*, and *sip-1* mRNA levels compared to controls. Moreover, feeding with 500 μM quercetin significantly increased *ctl-2*, *hsp-12.6*, *hsp-16.2*, and *hsp-70* expression compared to controls (Fig 5).

## Discussion

Recent studies have discussed the concepts of a 'health span' and 'lifespan' to differentiate between the period for which an individual remains healthy, versus the period between their birth and death. The *C. elegans* health span is represented by the time until worms begin to exhibit decreased motility in aged worms; therefore, the present study mainly examined the

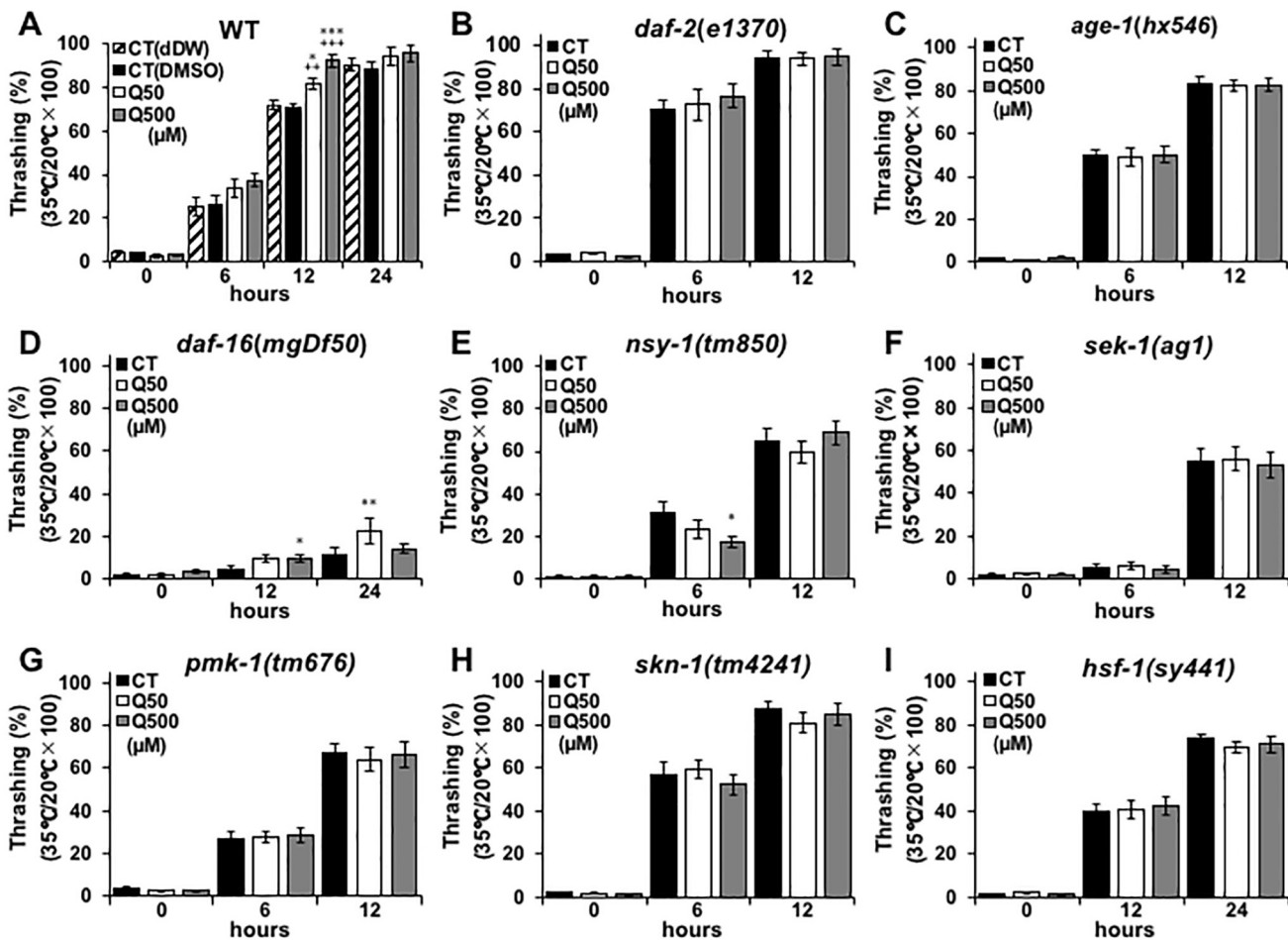

**Fig 3. Quercetin increased heat stress tolerance.** Movement counts were generated to assess the motility-recovery rate of **(A)** N2, **(B)** *e1370*, **(C)** *hx546*, **(D)** *mgDf50*, **(E)** *tm850*, **(F)** *ag1*, **(G)** *tm676*, **(H)** *tm4241*, and **(I)** *sy441 Caenorhabditis elegans* nematodes at **(A–C, E–H)** 6 h and **(D, I)** 12 h intervals after heat stress. Data are presented as the mean ± SEM, n = 10/group. +P < 0.01, +++P < 0.005 vs CT (dDW) and *P < 0.05, **P < 0.01 vs CT (DMSO) according to the conducted *Tukey's HSD*. All assays were conducted at least three times independently.

effects of quercetin on motility. No studies have previously evaluated quercetin in this context, and no previous reports have investigated the protective effects of quercetin on the motility of aged and stressed nematodes. Since severe heat stress disrupts nematode proteostasis (by causing aberrant protein folding and aggregation) in a manner similar to aging, the motility of heat-stressed worms is considered to be equivalent to that of aged worms, and to be representative of the nematode health span [33,34]. Thus, the present study investigated the physiological effects of quercetin on the motility of both aged and heat-stressed *C. elegans* nematodes.

In order to search similarity and differences between quercetin and other flavonoids, we choose epigallocatechin gallate (EGCG). EGCG didn't prolong the life span but enhance the heat stress tolerance of *C. elegans* [5]. In this study EGCG increased the motility of heat stressed- and aged nematodes similarly to quercetin (S1 and S2 Figs). These results suggested there are some similarity and differences of their physiological actions among flavonoids.

Consistent with previous studies, many genes were herein shown to adjust the nematode lifespan, including *daf-2*, *age-1*, *daf-16*, and *skn-1* [25,35,36]. Previous studies have also revealed that quercetin promotes longevity in nematodes by modulating the expression of *daf-*

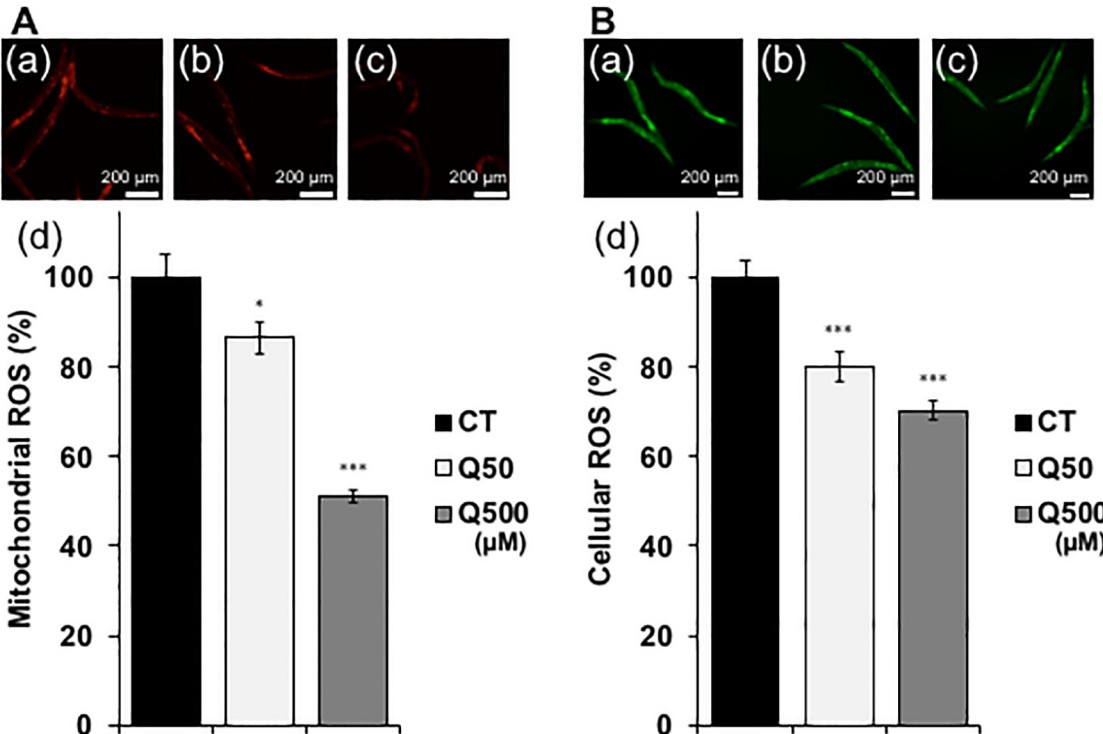

**Fig 4. Quercetin decreased mitochondrial- and intercellular ROS. (A)** Mitochondrial and **(B)** cellular ROS levels in **(a)** control (CT), and *Caenorhabditis elegans* nematodes treated with **(b)** Q50 mM, and **(c)** Q500 mM quercetin. **(d)** Analysis of data presented in **(a–c)**. Scale bars, 200 μm. Data are presented as the mean ± SEM. *P < 0.05, ***P < 0.005 according to the conducted *Tukey's HSD*. All assays were conducted at least three times independently.

*2*, *age-1*, *unc-43*, and *sek-1*, but not *daf-16* nor *skn-1* [16]. Notably, DAF-16 and SKN-1 are transcription factors that are controlled by the ILS and MAPK pathways, respectively [37,38], which themselves regulate the transcription of several genes related to stress tolerance [39]. In fact, DAF-16 is a FOXO homologue that has been shown to mediate both longevity and stress tolerance [36]. Similarly, SKN-1 is a homologue of Nrf2 that has been shown to contribute to phase-2 detoxification, and prolong lifespan [38]. The present study used *daf-2*, *age-1*, *daf-16*, *nsy-1*, *sek-1*, *pmk-1*, *skn-1*, and *hsf-1*-deficient mutants to assess whether the effects of quercetin on motility occur via the ILS and/or MAPK pathways. The transcription factor HSF-1 was also analysed, because it is known to mediate heat-stress tolerance, to interact with the ILS pathway, and to act with DAF-16 to drive the transcription of several heat-shock proteins [26,37,40]. Thus, the study analysed the interactions between quercetin, the ILS pathway, and HSF-1.

We analysed whether the concentration of DMSO used in this study affected to the lifespan, health span and heat stress tolerance of nematodes, since previous study showed DMSO shortened the lifespan of *C. elegans* [41]. Consequently, however, DMSO didn't have no influence on their lifespan, health span and heat stress tolerance (Figs 1, 2A and 3A). That's probably because in previous study the amount of DMSO was possibly much higher than that used in this study, since the authors dissolved DMSO in agar [41].

Quercetin feeding was found to prolong the lifespan of *C. elegans* (Fig 1), and suppress age-related motility retardation in the wild-type compared to the control nematodes (Fig 2A); however, this effect was not observed in the *daf-2(e1370)*, *age-1(hx546)*, *daf-16(mgDf50)*, *nsy-1* (*tm850*), *sek-1(ag1)*, *pmk-1(tm676)* nor the *skn-1(tm4241)* worms (Fig 2B–2H). Present studies

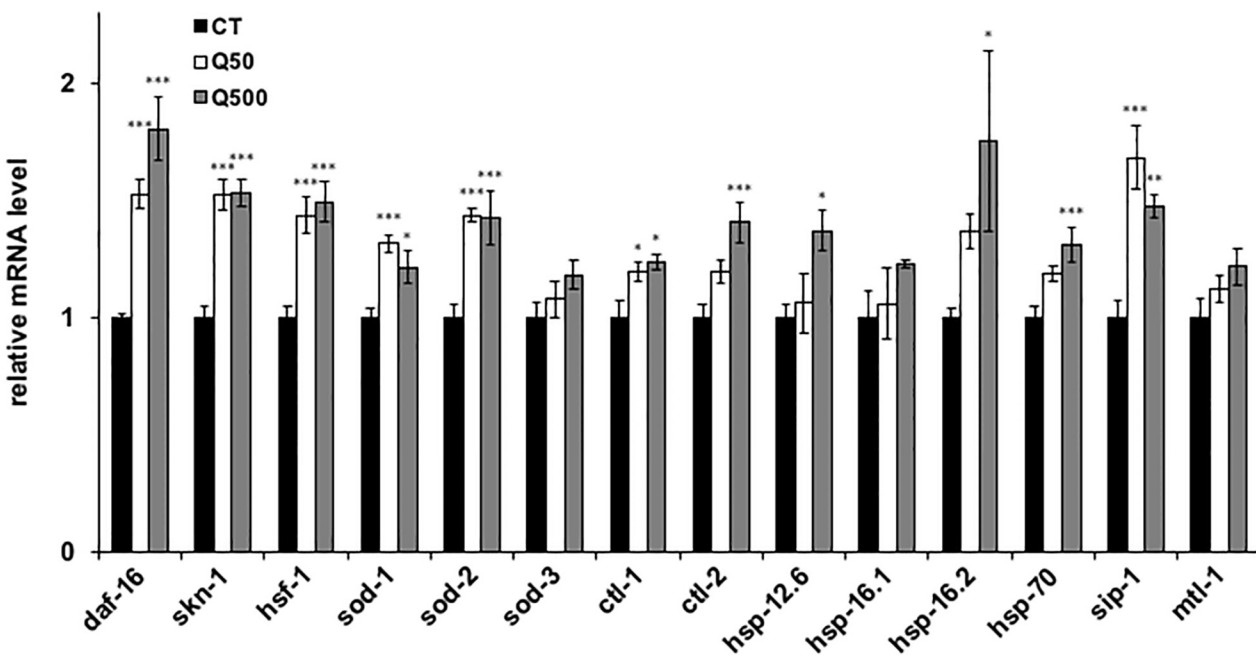

**Fig 5. Quercetin changed expression of genes.** The mRNA expression levels of the analysed genes are shown relative to that of the selected reference gene (*actin*). Data are presented as the mean ± SEM. $^*P < 0.05$, $^{***}P < 0.005$ according to the conducted *Tukey's HSD*. Assays were conducted at least three times independently.

have suggested that quercetin promotes motility in ageing nematodes via incurred effects on the ILS and MAPK pathway. Interestingly, previous studies have reported that neither *daf-16* nor *skn-1* mediate the quercetin-induced enhancement of nematode longevity [16]; however, in contrast, the results of the present suggest that quercetin increased the expression of both of these genes to enhance motility in aged worms, and thus increase their health span. Given that the phenotype of the aged *hsf-1(sy441)* worms was similar to that of the aged wild-type worms after quercetin feeding, it is unlikely that HSF-1 contributed to this phenotype (Fig 2I).

The improved recovery of the quercetin-fed nematodes from heat stress was suggested to occur via the upregulation of *daf-2*, *age-1*, *nsy-1*, *sek-1*, *pmk-1*, *skn-1*, *hsf-1*, and to a lesser extent, *daf-16* (Fig 3B–3I). Notably, the mRNA expression of *mtl-1* was not increased, despite the fact that this gene was previously shown to be associated with heat-stress tolerance, and to be a target of DAF-16 (Fig 5) [26]. In contrast, quercetin feeding induced increased transcription of the HSF-1 targets *hsp-16.2* and *hsp-70* (Fig 5) [25–27]. Together, these findings indicate that the effect of quercetin on heat-stress recovery was mediated predominantly and partially via the activation of HSF-1 and DAF-16, respectively, consistent with previous studies that showed that coactivating DAF-16 and HSF-1 significantly upregulates *hsp-12.6* and *sip-1* (but not *hsp-16.1*) mRNA expression (Fig 4) [26].

The present study further suggests that quercetin enhanced nematode motility after aging and heat stress via SKN-1 and the MAPK pathway. Interestingly, both have been shown to control the transcription of antioxidant enzymes, and thereby promote ROS scavenging [28]; thus, the observed improved nematode motility may have occurred because quercetin feeding induced the scavenging of ROS generated by senescence and heat stress. This hypothesis is supported by the fact that quercetin feeding upregulated transcription of the known ROS-scavenging SKN-1 target genes *sod-1* and *sod-2* (Fig 5). SOD-1 and SOD-2 have been shown to scavenge ROS in both the cytoplasm and mitochondria during nematode development [29],

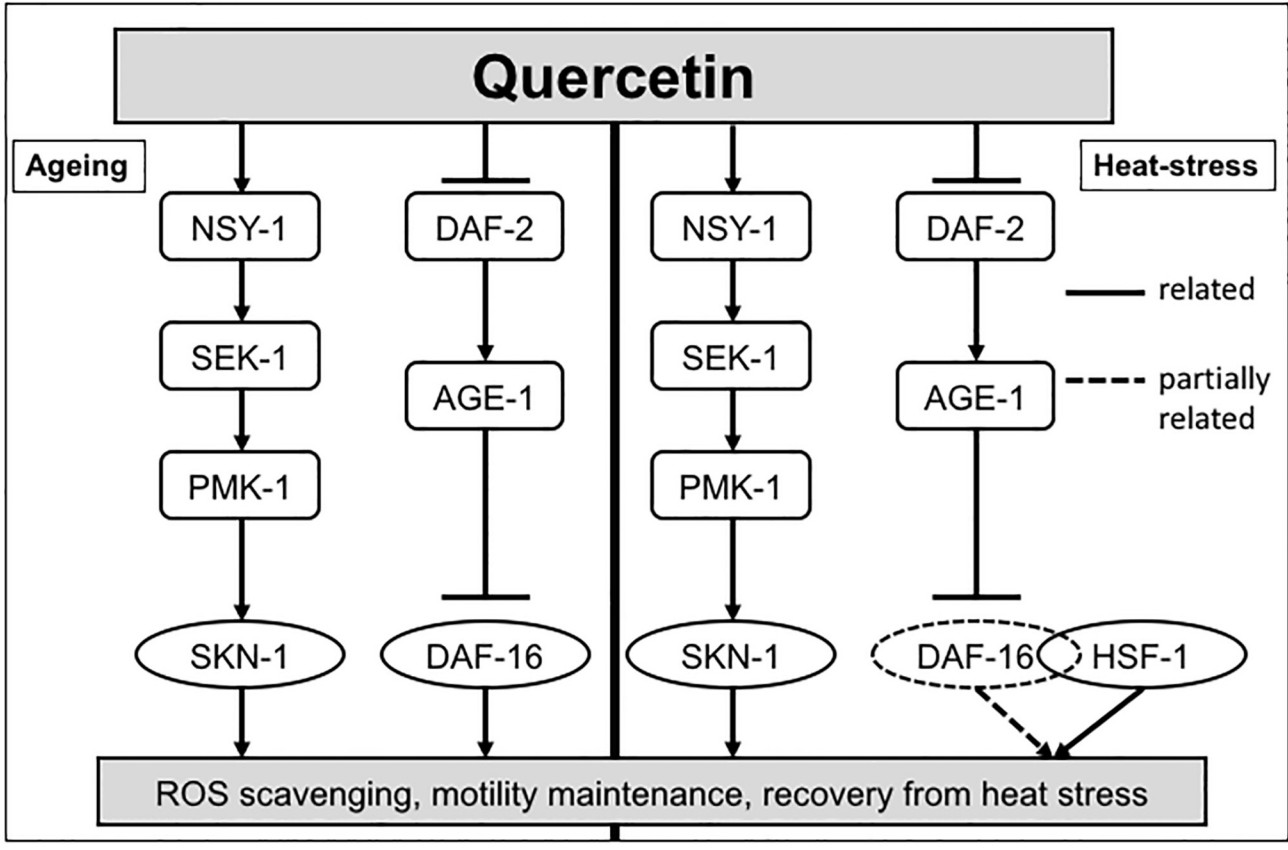

**Fig 6. Effects of quercetin on signalling pathways in *Caenorhabditis elegans*.** Signalling pathways shown to be modulated by quercetin in this study. Solid lines indicate relatedness, hashed lines indicate partial relatedness.

consistent with the fact that quercetin feeding herein decreased both cytoplasmic and mitochondrial ROS levels (Fig 4). We suggest that quercetin herein activated SOD-1 and SOD-2, and thereby caused ROS to be metabolized to $H_2O_2$ by SOD, and subsequently to $H_2O$ by catalase [40]. Furthermore, *ctl-1* and *ctl-2* are known to be regulated by DAF-16 [42]; thus, the herein observed upregulation of these genes suggests that DAF-16 indirectly contributed to ROS metabolism (Fig 5). Thus, we suggest that quercetin feeding induces ROS scavenging to both inhibit aging, and prolong the nematode health span.

The ILS pathway has been previously shown to regulate both DAF-16 and SKN-1 [35], consistent with the findings of the present study, which suggest that quercetin enhanced nematode motility after aging and heat stress by coactivating DAF-16 and SKN-1. Importantly, the present study revealed a novel quercetin-induced interaction between DAF-16 and HSF-1 (Fig 6); thus, it provides novel insights into the complex combination of signalling changes by which quercetin induces anti-aging effects [15]. Since these transcription factors are conserved in higher order animals, quercetin is a promising potential therapeutic target to improve and prolong the human health span.

## Supporting information

**S1 File.**
(DOCX)

**S1 Fig. Health span.** Movement counts were generated for N2 *Caenorhabditis elegans* nematodes every 3 days from day 0 until day 12, and presented relative to that calculated on day 0. Data are presented as the mean ± SEM, n = 10/group. $^*P < 0.05$, $^{***}P < 0.005$ vs CT according to the conducted *Tukey's HSD*. All assays were conducted at least three times independently. (TIFF)

**S2 Fig. Heat stress tolerance.** Movement counts were generated to assess the motility-recovery rate of N2 *Caenorhabditis elegans* nematodes at 6 h intervals after heat stress. Data are presented as the mean ± SEM, n = 10/group. $^*P < 0.05$, $^{***}P < 0.005$ vs CT (DMSO) according to the conducted *Tukey's HSD*. All assays were conducted at least three times independently. (TIFF)

**S3 Fig. Gene expression.** The mRNA expression levels of the analysed genes are shown relative to that of the quantity of actin in CT. Data are presented as the mean ± SEM. The quantities of genes in each condition are not significantly. Assays were conducted at least three times independently.
(TIFF)

## Acknowledgments

We thank CGC and NBRP for providing us nematodes. We appreciate to Suntory Global Innovation Center Limited for their help.

## Author Contributions

**Formal analysis:** Takaya Sugawara, Kazuichi Sakamoto.

**Investigation:** Takaya Sugawara.

**Methodology:** Takaya Sugawara.

**Project administration:** Kazuichi Sakamoto.

**Supervision:** Kazuichi Sakamoto.

**Validation:** Kazuichi Sakamoto.

**Writing – Review & Editing:** Kazuichi Sakamoto.

**Writing – original draft:** Takaya Sugawara.

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
