## [Decision Letter · Decision Letter 0]

15 Apr 2020

PONE-D-20-07154

Quercetin enhances motility in aged and heat-stressed Caenorhabditis elegans by modulating HSF-1 activity and insulin-like and p38-MAPK signaling

PLOS ONE

Dear Dr. Sakamoto,

Thank you for submitting your manuscript to PLOS ONE. After careful consideration, we feel that it has merit but does not fully meet PLOS ONE’s publication criteria as it currently stands. Therefore, we invite you to submit a revised version of the manuscript that addresses the points raised during the review process.

We would appreciate receiving your revised manuscript by May 30 2020 11:59PM. To enhance the reproducibility of your results, we recommend that if applicable you deposit your laboratory protocols in protocols.io, where a protocol can be assigned its own identifier (DOI) such that it can be cited independently in the future. For instructions see: http://journals.plos.org/plosone/s/submission-guidelines#loc-laboratory-protocols

We look forward to receiving your revised manuscript.

Kind regards,

Kyung-Jin Min

Academic Editor

PLOS ONE

Journal Requirements:

1. Thank you for including your funding statement within the acknowledgements section of your manuscript; "This work was supported in part by Grants-in-Aid for Scientific Research and Education from the University of Tsukuba, Japan. We appreciate Suntory Global Innovation Center Limited for their partial support."

"NO - Include this sentence at the end of your statement: The funders had no role in study design, data collection and analysis, decision to publish, or preparation of the manuscript."

Reviewers' comments:

Reviewer's Responses to Questions

**Comments to the Author**

1. Is the manuscript technically sound, and do the data support the conclusions?

Reviewer #1: Partly

Reviewer #2: Yes

2. Has the statistical analysis been performed appropriately and rigorously? 

Reviewer #1: Yes

Reviewer #2: Yes

3. Have the authors made all data underlying the findings in their manuscript fully available?

Reviewer #1: Yes

Reviewer #2: Yes

4. Is the manuscript presented in an intelligible fashion and written in standard English?

Reviewer #1: Yes

Reviewer #2: Yes

5. Review Comments to the Author

Reviewer #1: 1. Line 69: Please delete the word of“signaling”at the period

2. Please describe in detail of OP or Q plates

3. Please point the detail concentration of OP and O in each experiment

4. Line 224-227: the authors showed that the heat stress disrupts nematode proteostasis by causing aberrant protein folding and aggregation in a manner similar to aging, so the motility of heat-stressed worms is considered to be equivalent to that of aged worms, and to be representative of the nematode healthspan. Please list related references

5. It is recommended offer the data of the positive control in each experiment

Reviewer #2: Obtained data on quercetin are not consistent with the results of other model studies (https://www.ncbi.nlm.nih.gov/pmc/articles/PMC5179547/) and there is not enough discussion of the reasons for such differences.

Only one house-keeping gene as a reference in RT-PCR, but actine has age-related changes in its expression. It is better to try 2-3 different house-keeping genes.

The authors have to carefully check the style. For example the word "modulate" used 4 times in the abstract.

In a row 69: signaling [4,14-16]. signaling

6. PLOS authors have the option to publish the peer review history of their article (what does this mean?). If published, this will include your full peer review and any attached files.

Reviewer #1: No

Reviewer #2: No

---

## [Author Response · Author response to Decision Letter 0]

3 Aug 2020

Replies to reviewers' comments:

*All corrections are shown in red in the text.

Reviewer #1

1. Line 69: Please delete the word of “signaling” at the period

<Reply 1>

We appreciate to the reviewers’ comment. Following the reviewers’ suggestion, we deleted the word of “signaling”

See -� P4, L69 

2. Please describe in detail of OP or Q plates

<Reply 2>

OP plate and Q plate are the plates coated with E. coli (OP50) and quercetin (50µl, 500µl), respectively, onto a NGM plate as written in Materials and methods.

See -� P5, L80

3. Please point the detail concentration of OP and O in each experiment

<Reply 3>

Basically we used the concentration 50µl and 500µl only. We described the concentration in Materials and methods.

 See -� P7, L96 and 107, P8, L122, P9, L134 and 142, P10, L150

4. Line 224-227: the authors showed that the heat stress disrupts nematode proteostasis by causing aberrant protein folding and aggregation in a manner similar to aging, so the motility of heat-stressed worms is considered to be equivalent to that of aged worms, and to be representative of the nematode healthspan. Please list related references

<Reply 4>

We appreciate to the reviewers’ comment. Following the reviewers’ suggestion, we inserted two references [33, 34].

[33] Ravi Raghav Sonani. et al. Phycoerythrin extends life span and health span of Caenorhabditis elegans. 

Age (Dordr). 2014;36(5):9717. doi: 10.1007/s11357-014-9717-1.

[34] Philippe Verbeke, et al. Heat Shock Response and Ageing: Mechanisms and Applications.

Cell Biology International 2001, Vol. 25, No. 9, 845–857.

See -� P15, L239

5. It is recommended offer the data of the positive control in each experiment

<Reply 5>

We appreciate to the reviewers’ comment. Following the reviewers’ suggestion, we inserted one reference [5] and supplemental data using EGCG, as a positive control.

[5] Zhang, L. et al. Significant longevity-extending effects of EGCG on Caenorhabditis elegans under stress. Free Radical Biology and Medicine, 46(3), 414-421 (2009).

 See -� P3, L42, P12, L185, P13, L202, P15, L241 and in supplemental data

Reviewer #2

Obtained data on quercetin are not consistent with the results of other model studies (https://www.ncbi.nlm.nih.gov/pmc/articles/PMC5179547/) and there is not enough discussion of the reasons for such differences.

<Reply 1>

As written in the previous paper [13], low concentration of quercetin has positive effects, whereas high concentration has negative effects. The effects on aging phenotypes might be different depending on the concentration of quercetin and animals to apply.

[13] Proshkina, E. et al. Geroprotective and Radioprotective Activity of Quercetin, (-)-Epicatechin, and Ibuprofen in Drosophila melanogaster. Front. Pharmacol. 7, 505 (2016).

 See -� P3, L46

Only one house-keeping gene as a reference in RT-PCR, but actin has age-related changes in its expression. It is better to try 2-3 different house-keeping genes.

<Reply 2>

We performed qPCR assay using tba-1 and pmp-3 as the reference genes recommended in a paper [31], and inserted these data and one reference [31] into text. Actin could be a proper reference gene because the quantity of mRNA expression of actin, tba-1 and pmp-3 didn’t change significantly.

 [31] Zhang, Y et al. Selection of reliable reference genes in Caenorhabditis elegans for analysis of nanotoxicity. PloS one 7.3 (2012): e31849.

 See -� P10, L156, P14, L219 and in supplemental data

The authors have to carefully check the style. For example the word "modulate" used 4 times in the abstract.

In a row 69: signaling [4,14-16]. Signaling

<Reply 3>

We appreciate to the reviewers’ comment. Following the reviewers’ suggestion, we replaced the words “modulate” and “signalling” to some others or removed and abbreviate insulin-like signaling as ILS.

 See -� P2, L28-35, P4, L68, P5, Table1, P16, L250, 256-259, P17, L271 and 

273 and P19, L305

---

## [Decision Letter · Decision Letter 1]

19 Aug 2020

Quercetin enhances motility in aged and heat-stressed Caenorhabditis elegans nematodes by modulating both HSF-1 activity, and insulin-like and p38-MAPK signalling

PONE-D-20-07154R1

Dear Dr. Sakamoto,

We’re pleased to inform you that your manuscript has been judged scientifically suitable for publication and will be formally accepted for publication once it meets all outstanding technical requirements.

Kind regards,

Kyung-Jin Min

Academic Editor

PLOS ONE

Additional Editor Comments (optional):

Reviewers' comments:

Reviewer's Responses to Questions

**Comments to the Author**

1. If the authors have adequately addressed your comments raised in a previous round of review and you feel that this manuscript is now acceptable for publication, you may indicate that here to bypass the “Comments to the Author” section, enter your conflict of interest statement in the “Confidential to Editor” section, and submit your "Accept" recommendation.

Reviewer #1: All comments have been addressed

Reviewer #3: All comments have been addressed

2. Is the manuscript technically sound, and do the data support the conclusions?

Reviewer #1: Yes

Reviewer #3: Yes

3. Has the statistical analysis been performed appropriately and rigorously? 

Reviewer #1: Yes

Reviewer #3: Yes

4. Have the authors made all data underlying the findings in their manuscript fully available?

Reviewer #1: Yes

Reviewer #3: Yes

5. Is the manuscript presented in an intelligible fashion and written in standard English?

Reviewer #1: Yes

Reviewer #3: Yes

6. Review Comments to the Author

Reviewer #1: This paper deals with the movement ability of Quercetin enhances heat-stressed in C. elegans. The subject and contents are I think good for this journal.

Reviewer #3: (No Response)

7. PLOS authors have the option to publish the peer review history of their article (what does this mean?). If published, this will include your full peer review and any attached files.

Reviewer #1: No

Reviewer #3: **Yes: **Kyung-Jin Min

---

## [Editor Report · Acceptance letter]

21 Aug 2020

PONE-D-20-07154R1 

 Quercetin enhances motility in aged and heat-stressed *Caenorhabditis elegans* nematodes by modulating both HSF-1 activity, and insulin-like and p38-MAPK signalling 

Dear Dr. Sakamoto:

I'm pleased to inform you that your manuscript has been deemed suitable for publication in PLOS ONE. Congratulations! Your manuscript is now with our production department. 

Kind regards, 

on behalf of

Dr Kyung-Jin Min 

Academic Editor

PLOS ONE